# A randomised survey of the quality of antibiotics and other essential medicines in Indonesia, with volume-adjusted estimates of the prevalence of substandard medicines

**Elizabeth Pisani** *, **Ayu Rahmawati, Esti Mulatsari, Mawaddati Rahmi**, **William Nathanial, Yusi Anggriani, on behalf of the STARmeds Study Group**[¶]

Faculty of Pharmacy, Universitas Pancasila, Lenteng Agung, Jakarta Selatan, Indonesia

[¶] Membership of the STARmeds Study Group is provided in the Acknowledgments.
* pisani@ternyata.org

**Data Availability Statement:** Additional data are available in three locations, all within the

## Abstract

The World Health Organization warns that substandard and falsified medicines threaten public health in low- and middle-income countries. However, medicine quality surveys are often small and unrepresentative of the market, and the true scale of the problem remains unknown. We conducted a large field survey of essential medicines in Indonesia, and investigated how weighting survey results by market volume altered estimates of medicine quality. We collected 1274 samples of allopurinol, amlodipine, cefixime, amoxicillin and dexamethasone from the internet and a randomised sample of all outlet-types where medicines are sold or dispensed in seven districts across the world's fourth most populous nation. We conducted compendial testing for identity, assay, dissolution and uniformity. Samples that failed any chemical test were considered substandard. We compared raw prevalence of substandard medicines with prevalence adjusted by the national sales volume of each brand, relative to its weight among survey samples. The weighted prevalence of substandard products was 4.4%, 47% lower than the raw estimate (8.2%). Only 0.5% of samples (unweighted 1.2%) deviated from permitted limits by more than 10%. More antibiotics failed testing than other medicines (weighted prevalence 8.5 vs 3.1; raw prevalence 13.6 vs 4.9, both p<0.000). We found no relationship between quality and price; branded status; public procurement status; or outlet type. In Indonesia, unweighted survey data appeared to substantially over-estimate the health threat posed by substandard or falsified medicines. The types of sampling bias that led to over-representation of poor quality products in our survey are also indicated in other published surveys, possibly exaggerating the scale of the threat to public health posed by substandard and falsified medicines globally. Weighting survey results by sales volume likely improves robustness of estimates of medicine quality measured in field surveys.

STARmeds repository, and identified by DOIs in the text or references. Code and supplementary data for this specific paper (including the weighting and analysis code in Stata format, the supplementary methods description, supplementary tables and author contributions) are at: https://doi.org/10.7910/DVN/QRKDWG Data and documentation related to STARmeds fieldwork more generally are in the study archive. This archive is easiest to use in Tree view. It contains the sample level data produced by the STARmeds field study, including all of the raw laboratory data, in csv format. Also included are laboratory protocols and a more detailed description of methods. The archive can be accessed at: https://doi.org/10.7910/DVN/RKYICP. Finally, we provide a free toolkit to help researchers and regulators design and implement medicine quality field surveys using mystery shoppers. The toolkit contains downloadable and adaptable versions of data collection software, field control forms, field worker contracts and other potentially useful documentation. The Toolkit can be downloaded from: https://doi.org/10.7910/DVN/OBIDHJ.

**Funding:** The study was funded by UK taxpayers through the UK Department of Health and Social Care and the National Institute for Health Research, under NIHR Global Health Policy and Systems Research Commissioned Awards (https://www.nihr.ac.uk/), grant number NIHR131145. The funders had no role in study design, data collection and analysis, decision to publish, or preparation of the manuscript.

**Competing interests:** Yusi Anggriani is a member of the Indonesian Ministry of Health's advisory committee on medicine pricing, and a member of the World Health Organization Technical Advisory Group on Pricing Policies for Medicines. Elizabeth Pisani has worked as a consultant on research commissioned by the WHO Incidents and Substandard/Falsified medical products team. All other authors report no conflict of interest. This does not alter our adherence to PLOS ONE policies on sharing data and materials.

## Introduction

In 2017, the World Health Organization (WHO) estimated that 10.5% of medicines in all low- and middle-income countries were substandard (they did not meet the standards laid out in their market authorisation paperwork) or falsified (they deliberately misrepresented content, identity or source) [1]. If correct, this figure should be alarming to those striving to expand access to medicines for 7.3 billion citizens of these countries.

The 2017 estimate was based on a review of 100 studies, mostly of malaria or TB medicines sampled from retail outlets in sub-Saharan Africa (see study archive for extracted data [2]). More recent reviews have found similar results, with overall aggregate prevalences of poor quality antibiotic, cardiovascular and diabetes medicines in low- and middle-income countries of 17.4, 15.4, and 10.8% respectively [3–6]. All note that prevalence reported in published studies are unlikely to be generalisable because of small sample sizes, unrepresentative study designs and variations in medicines included; tests performed; and pharmacopeia and definitions used. No reviewed study considered market share of different brands when calculating results.

Global health actors continue to call for survey data that would provide a more nuanced understanding of the actual prevalence and distribution of poor quality medicines circulating in specific markets [7, 8]. But the cost of testing medicines, combined with methodological challenges, mean truly representative sampling is rarely feasible [9, 10].

The STARmeds study, reported here, developed sampling and weighted estimation methods designed to increase the representativeness of medicine quality survey data in resource-limited settings.

## Materials and methods

All methods are described in greater detail in S1 Methods, according to MEDQUARG guidelines [10]. That document provides details of secondary data sources, medicine and site selection, sample size calculation, sample frame construction, sample collection and handling, data entry and management, laboratory testing, ethics protections, and estimation methods. We summarise these briefly here. Additional data, including a MEDQUARG reporting checklist, estimation code, laboratory protocols, full sample-level data and codebook, and a free, downloadable toolkit for conducting medicine quality surveys are provided in the study archive [11, 12].

### Study setting

Middle-income Indonesia is home to one of the world's largest and most generous public health insurance schemes; it also has a vibrant domestic pharmaceutical sector. National health insurance now covers 249 million people [13]. Since inception of the system in 2014, national consolidated tenders pushed down the price of most medicines while increasing volumes consumed [14]. Medicines are provided free to insured patients, but administrative procedures are burdensome and many patients buy medicines outside the public system. In a nationally-representative survey of 345,000 households conducted in 2023, 80% of people who reported any symptoms of illness in the preceding month said they had self-treated. Only 35% of those with symptoms visited a health facility. Medicines accounted for just under a fifth of all out-of-pocket spending on health. Most of this spending was on modern rather than traditional medicines (42% on over-the-counter medicines and 39% on medicines bought with a prescription) acquired directly from health personnel [15]. Public procurement accounted for just 14.4% of the estimated US$3.6 billion spent on medicines in Indonesia in 2022 [16, 17].

Indonesian manufacturers warned falling prices may threaten medicine quality [18–20]. Cases of falsified vaccines in 2017 and contaminated paediatric syrups in 2022 added to public mistrust of medicine quality in Indonesia [21, 22]. However, in an independent field survey of cardiovascular/diabetes medicines collected from regulated and unregulated outlets in one district in 2021, all 204 samples passed testing for assay (which measures the percent of labelled active ingredient identified in the sample) and dissolution (the percent of active ingredient dissolved within a specified timeframe) [23].

Indonesia's medicine regulator (*Badan Pengawas Obat dan Makanan*, BPOM) conducts extensive proactive post-market surveillance, though only from regulated outlets. BPOM tested 10,980 medicines collected through randomised sampling in the regulated supply chain in 2021. 3.7% were out-of-specification (*tidak memenuhi syarat*), meaning they failed at least one pharmacopeial test, were expired or damaged at the time of sampling, or were incorrectly registered or labelled. Another 2,559 samples were collected using risk-based sampling, focused on products of public health importance with high potential for quality defects; 5.2% failed on at least one dimension, for an overall estimate of 4.0% out-of-specification products nationally [24, 25].

## Study design

STARmeds medicines were chosen based on public health importance, diversity of suppliers, risk of falsification and feasibility, in consultation with BPOM, the Ministry of Health, the national health insurer, the association of hospitals, and other stakeholders. In determining public health importance we considered the prevalence of the conditions treated by the medicine; the volumes procured in the public system and sold in pharmacies nationally; and the financial burden on the public health insurance system. See S1 Methods for details.

Included products are shown in Table 1; all require prescriptions in Indonesia.

Sampling locations were chosen purposively to reflect Indonesia's geographic and economic diversity. We included all outlet types from which Indonesian patients commonly acquire medicines. Regulated outlets included pharmacies; public and private hospitals; primary health centres, and licensed phone apps. Unregulated outlets (technically forbidden from dispensing prescription medicines) included over-the-counter medicine shops, individual shops in bulk medicine markets; doctors; midwives and internet platforms. In each area, we listed and verified all outlets by type. In consultation with experts from Statistics Indonesia, we drew up a sample frame which took into account the relative size of each sampling location, and specified the desired number of sampling venues for each district and outlet type. We then selected target outlets randomly from the list for each location and outlet type, using Stata 17's random generation function.

**Table 1. Products included in the STARmeds study.**

| Active ingredient | Primary use | Target doses | # of registered products* | Falsification risk |
|---|---|---|---|---|
| Allopurinol | Anti-hyperuricemia (Gout) | 100mg tablet; | 65 | Used non-medically |
| | | 300mg tablet | 46 | |
| Amlodipine | Anti-hypertensive | 5mg tablet | 112 | None |
| Amoxicillin | Antibiotic | 500mg tablet/capsule; | 85 | None |
| | | 125mg dry syrup | 68 | |
| Dexamethasone | Anti-inflammatory | 0.5mg tablet | 59 | None |
| Cefixime | Antibiotic | 100mg tablet/capsule | 45 | Not free at primary level; relatively expensive |

*Number of different brands/branded generics of the target doses and formulations registered in the Indonesian market, from the public domain BPOM product registration database (2022)

**Table 2. Sampling locations, characteristics and sampling dates.**

| District/ sampling area | Geographic area | Population per km/ sq.* | Annual per capita GDP (US $)** | Outlet randomisation method | Sampling dates, 2022 |
|---|---|---|---|---|---|
| Greater Jakarta | Central megacity | 14,792 | 7,784 | Two-stage PPS | 15–20 February |
| Surabaya city | Large city | 8,225 | 15,270 | SRS | 1–5 March |
| Malang regency | Semi-rural | 733 | 2,941 | One stage PPS | 1–5 March |
| Medan city | Large city | 8,525 | 7,553 | SRS | 22–26 March |
| Labuhan Batu regency | Remote rural | 225 | 5,534 | Take all for pharmacies and SRS for other outlets | 22–26 March |
| Kupang City | Small city | 2,335 | 3,784 | SRS | 29 March-2 April |
| Timor Tengah Selatan regency | Remote rural | 118 | 1,289 | Take all for pharmacies and SRS for other outlets | 5–8 April |

PPS: Probability proportionate to size. SRS: Simple random sampling.

*2022 data from BPS/StatisticsIndonesia

** 2022 data from BPS/StatisticsIndonesia. Rupiah values converted at Bank Indonesia average rate for 2022: 1 USD = 14,870.61 rupiah; Greater Jakarta is weighted average for sampling districts.

All pharmacies and medicine shops were privately owned; numbers depended on the location. We selected a district public hospital (in most districts there is only one), one private hospital, and one public primary health centre (*puskesmas*) in each sampling area. All doctors and midwives included were in private practice. We randomly chose a total of five in each location. Further details are provided in S1 Methods. Table 2 shows locations, characteristics, sampling dates and type of outlet randomisation.

## Sample collection and handling

Trained mystery shoppers bought medicines from selected retail outlets, using individualised sample frames which specified medicines, doses and a price point (cheaper or more expensive) but not a brand. Shoppers carried prescriptions, offering them if requested. In health facilities, study staff sampled overtly, buying a branded and an unbranded version of each study medicine if available.

Each sample was stored in a separate, pre-barcoded bag. Field data (barcode, geolocation, price) were entered on smartphones, using KoboCollect software [26]. Samples were delivered daily to local study hubs. They were inspected for anomalies in packaging or labelling. Data entry staff entered additional product details and took high-resolution photographs using tablets pre-loaded with the study software. Samples were stored with a temperature logger in an airconditioned room until transfer to the laboratory.

## Laboratory testing

Samples were tested using United States Pharmacopeia (USP) reference standards, at PT Equilab International in Jakarta. For cefixime capsules, we followed Supplement 1 of *Farmakope Indonesia* 6<sup>th</sup> [27]. All other medicines were tested according to USP 43, NF38 monographs [28]. The acceptance criteria for each product are shown in Table 3.

Identification, assay and uniformity of assay were tested using high performance liquid chromatography (HPLC-UV Waters, Alliance 2695 with UV Detector 2489). HPLC was also used in dissolution testing for amoxicillin tablets and dexamethasone. Dissolution of amoxicillin capsules, allopurinol, amlodipine and cefixime was analysed using Spectrophotometer-UV/

**Table 3. Acceptability criteria for pharmacopeial tests, USP 43 NF38.**

| Active ingredient | Identification | Assay (%) | Dissolution (%) ('Q') | Content Uniformity |
|---|---|---|---|---|
| Allopurinol | Retention time of the major peak of the sample solution corresponds to that of the reference solution | 93.0–107.0 | 75% in 45 minutes | NA |
| Amoxicillin, tablet | | 90.0–120.0 | 75% in 30 minutes | NA |
| Amoxicillin, capsule | | 90.0–120.0 | 80% in 60 minutes | NA |
| Amoxicillin, dry syrup | | 90.0–120.0 | NA | NA |
| Cefixime, tablet | | 90.0–110.0 | 75% in 45 minutes | NA |
| Cefixime, capsule | | 90.0–110.0 | 80% in 45 minutes | NA |
| Amlodipine | | 90.0–110.0 | 75% in 30 minutes | Acceptance value ≤ 15.0, and no individual tablet has an assay value that falls outside USP-specified limits. |
| Dexamethasone | | 90.0–110.0 | 80% in 30 minutes | |

NA: Not applicable

VIS (Shimadzu UV-1800). No dissolution testing was performed on amoxicillin dry syrup formulation. Full laboratory protocols are in the study archive.

We could not afford to test for impurities.

## Product verification and falsification

We provided all 79 market authorisation holders with per-sample data and high-definition photos of primary and (if available) secondary packaging, asking them to verify all sampled batch numbers, expiry dates and maximum retail prices against their production records. Samples with confirmed anomalies were considered falsified, as were any with no or incorrect active ingredient.

## Analysis and estimation

We used Stata 17 and 18 for data cleaning, weighting and analysis [29].

## Product quality

We defined a substandard sample as one that failed any pharmacopeial test, using the limits shown in Table 3.

Raw prevalence was calculated by dividing the number of substandard samples by the number tested.

Previously-suggested measures of "extreme deviation" do not take into account the variations in assay values permitted by most pharmacopeia [30, 31]. We calculated a single measure of deviation from permitted values as the maximum among the following:

percentage points by which assay exceeds or falls short of the limits of acceptability;

percentage points by which final dissolution value falls below the permitted threshold.

To distinguish between marginal and more extreme deviation, we also report prevalence of samples that deviate from permitted levels by over 10%.

### Weighted estimates

Outlets were chosen randomly, but mystery shoppers could not credibly request a list of individual medicines by volume for truly representative sampling. In our adjusted estimates, we thus weighted each sample in our study by the ratio of a brand, dose and formulation's market share to its weight among our study samples, using the total market for all five medicines as the universe, as follows:

$$\frac{\left(\frac{market\_volume\_BrandA}{\sum market\_volume\_all\_study\_medicines}\right)}{\left(\frac{\sum samples\_BrandA}{\sum all\_samples}\right)}$$

For estimates of prevalence by medicine, we recalculated the relative volumes within the universe of each active ingredient. Details are provided in S1 Methods.

### Volume by product (active ingredient, formulation, brand and dose)

For calendar 2022, pharmaceutical data aggregator IQVIA listed non-zero sales volumes in the Indonesian market for 467 separate versions of the medicines and formulations included in the study. Each was a unique combination of medicine, formulation, dose and brand (or, for unbranded generics, market authorisation holder). We used these volumes as the basis for our estimates. For public procurement products (n = 17), we added national public transaction volumes for calendar 2021 to IQVIA volumes, adjusting to avoid double counting in the hospital sector. We redistributed IQVIA's unnamed "generic manufacturer" volumes across unbranded generics as described in S1 Methods, which also explains how we imputed volumes for 18 products collected in our survey but missing from IQVIA data.

### Price-related analyses

When investigating the relationship between price and quality, we used the price paid for the specific sample, per smallest counting unit. Samples of public procurement medicines acquired at no cost were priced at the public procurement cost plus a government-permitted margin of 28% for tax and handling.

### Ethics and reporting

The study design was widely discussed with BPOM and a multisectoral national working group on medicine quality. The study protocol was approved by institutional review boards at Universitas Indonesia (970/UN2.F1/ETIK/PPM.00.02/2020) and Imperial College London (21IC7265). We also obtained written permissions from the health department or other competent authority in each sampling district.

Dedicated study staff in the district research hubs provided full-time problem-solving support by phone to mystery shoppers. We immediately notified BPOM of any suspect product; we shared sample-level results for all study products once certificates of analysis were issued.

## Results

We collected and tested 1,274 samples, as shown in Table 4; 82 were not of targeted doses. Four locally-made branded products were registered by multinational companies. The rest were registered by 75 different Indonesian pharmaceutical firms; medicines were made by 72 different manufacturers, all in Indonesia. (This excludes two internet-acquired samples of locally registered brands apparently packed for and imported from other markets.) We

Table 4. Number of samples tested by location of collection, medicines and dose, and number of unique products collected by medicine, dose and branded status.

| Medicine & dose | Greater Jakarta | North Sumatra Rural | North Sumatra Urban | East Java Rural | East Java Urban | NTT Rural | NTT Urban | Online | Total | API Total | Branded | Unbranded | Unique products |
|---|---|---|---|---|---|---|---|---|---|---|---|---|---|
| Allopurinol 100mg | 43 | 14 | 26 | 20 | 31 | 12 | 19 | 55 | 220 | | 19 | 16 | 35 |
| Allopurinol 300mg | 21 | 0 | 12 | 3 | 11 | 2 | 4 | 24 | 77 | 297 | 17 | 5 | 22 |
| Amlodipine 10mg | 3 | 0 | 4 | 0 | 4 | 1 | 5 | 15 | 32 | | 8 | 13 | 21 |
| Amlodipine 5mg | 55 | 13 | 25 | 18 | 27 | 7 | 21 | 46 | 212 | 244 | 28 | 27 | 55 |
| Amoxicillin 500mg | 44 | 11 | 29 | 22 | 28 | 11 | 22 | 57 | 224 | | 32 | 7 | 39 |
| Amoxicillin, dry syrup | 22 | 5 | 17 | 2 | 9 | 2 | 8 | 10 | 75 | 299 | 15 | 9 | 24 |
| Cefixime 100mg | 38 | 9 | 20 | 15 | 20 | 10 | 15 | 41 | 168 | | 21 | 7 | 28 |
| Cefixime 200mg | 7 | 2 | 3 | 1 | 0 | 1 | 2 | 11 | 27 | 195 | 6 | 7 | 13 |
| Dexamethasone 0.5mg | 42 | 12 | 27 | 19 | 34 | 10 | 18 | 58 | 220 | | 25 | 10 | 35 |
| Dexamethasone 0.75mg | 4 | 1 | 2 | 2 | 0 | 0 | 0 | 10 | 19 | 239 | 8 | 0 | 8 |
| **Total** | **279** | **67** | **165** | **102** | **164** | **56** | **114** | **327** | **1,274** | | **179\*** | **101\*\*** | **280** |

API: Active pharmaceutical ingredient

*Excludes two illegally imported versions of locally registered products

**Unbranded generics from the same market authorisation holder are counted separately for each active ingredient and dose/formulation

collected one originator product, 178 unique medicines sold under proprietary brand names (branded generics), and 101 unique unbranded products, differentiated by market authorisation holder. Table 4 shows the distribution of tested samples by sampling area and medicine, and the number of unique products. Table 5, meanwhile, shows the distribution of samples by location and type of outlet.

Table 5. Distribution of samples and unique products by outlet type and location.

| | | Pharmacy Samples | Pharmacy Products | OTC medicine shop Samples | OTC medicine shop Products | Primary health centre Samples | Primary health centre Products | Hospital Samples | Hospital Products | Doctor Samples | Doctor Products | Midwife Samples | Midwife Products |
|---|---|---|---|---|---|---|---|---|---|---|---|---|---|
| Greater Jakarta | | 194 | 90 | 48 | 44 | 6 | 6 | 21 | 18 | 8 | 7 | 2 | 2 |
| North Sumatra | Medan | 125 | 79 | 1 | 1 | 5 | 5 | 10 | 10 | 14 | 11 | 10 | 10 |
| | Labuhan Batu | 25 | 23 | 3 | 3 | 6 | 6 | 12 | 12 | 10 | 10 | 11 | 11 |
| East Java | Surabaya | 118 | 55 | 0 | 0 | 2 | 2 | 21 | 15 | 13 | 12 | 10 | 10 |
| | Malang district | 69 | 51 | 0 | 0 | 5 | 5 | 14 | 14 | 5 | 5 | 9 | 9 |
| NTT | Kupang city | 65 | 40 | 0 | 0 | 9 | 7 | 21 | 16 | 19 | 17 | 0 | 0 |
| | TTS | 36 | 22 | 0 | 0 | 4 | 4 | 14 | 11 | 0 | 0 | 2 | 2 |
| **Total** | | **632** | 165 | **52** | 48 | **37** | 25 | **113** | 54 | **69** | 49 | **44** | 34 |

**Online**

| Regulated Geo-positioned app Samples | Regulated Geo-positioned app Products | Regulated Registered online medicine sales site Samples | Regulated Registered online medicine sales site Products | Semi-regulated On-line sales from verified pharmacy Samples | Semi-regulated On-line sales from verified pharmacy Products | Unregulated Samples | Unregulated Products | Total Samples | Total Products |
|---|---|---|---|---|---|---|---|---|---|
| 80 | 62 | 17 | 14 | 44 | 37 | 186 | 123 | 327 | 177 |

OTC: Over the counter

NTT: Nusa Tenggara Timur; TTS: Timor Tengah Selatan

**Table 6. Pharmacopeial test results by medicine type.**

| Medicine & formulation | Assay | | | Dissolution | | | Uniformity of assay | | | Total | | | Confirmed falsified* | |
|---|---|---|---|---|---|---|---|---|---|---|---|---|---|---|
| | N | Fail | % Fail | N | Fail | % Fail | N | Fail | % Fail | N | Fail | % Fail | N | Yes |
| Allopurinol tablets | 297 | 13 | **4.4** | 283 | 8 | **2.8** | 0 | - | - | **297** | **18** | **6.1** | 111 | 5 |
| Amlodipine tablets | 244 | 0 | **0.0** | 236 | 0 | **0.0** | 83 | 4 | **4.8** | **244** | **4** | **1.6** | 92 | 2 |
| Amoxicillin capsules | 36 | 0 | **0.0** | 32 | 0 | **0.0** | 0 | - | - | **36** | **0** | **0.0** | 7 | 0 |
| Amoxicillin dry syrup | 75 | 15 | **20.0** | 0 | - | - | 0 | - | - | **75** | **15** | **20.0** | 31 | 0 |
| Amoxicillin, tablets | 188 | 3 | **1.6** | 187 | 12 | **6.4** | 0 | - | - | **188** | **15** | **8.0** | 95 | 4 |
| Cefixime capsules | 180 | 11 | **6.1** | 167 | 36 | **21.6** | 0 | - | - | **180** | **32** | **17.8** | 106 | 7 |
| Cefixime tablets | 15 | 4 | **26.7** | 7 | 1 | **14.3** | 0 | - | - | **15** | **5** | **33.3** | 6 | 2 |
| Dexamethasone tablets | 239 | 11 | **4.6** | 228 | 9 | **3.9** | 93 | 2 | **2.2** | **239** | **16** | **6.7** | 130 | 1 |
| **All** | **1274** | **57** | **4.5** | **1140** | **66** | **5.8** | **176** | **6** | **3.4** | **1274** | **105** | **8.2** | **578** | **21** |

*A total of 44 Market Authorisation holders provided confirmation data for 578 samples. These included 27 which were not laboratory tested because of budget limitations (see S1 Methods). None of the latter were reported to be falsified.

Note: Sample level data, with granular pharmacopeial test results, can be downloaded from the study archive for more detailed analysis [11].

Branded medicines dominated the online samples (72%); in physical outlets, 51% of samples were branded.

## Raw quality estimates

All samples contained the correct active ingredient. Table 6 shows further test results, and the number confirmed falsified among the 578 samples for which we received confirmation from market authorisation holders.

Overall, 8.2% of samples failed at least one pharmacopeial test, with significant differences by medicine type. Samples expired at most recent testing date (n = 47) were no more likely to fail testing than unexpired samples (8.5% vs 8.2%, p = 0.95). The anti-hypertensive medicine amlodipine, the only chronic disease medicine in the study, had the lowest testing failure rate, at 1.6% (all failing in uniformity of content only). The antibiotics amoxicillin and cefixime had the highest testing failure rates (10.0% and 19.0% respectively, totalling 13.6%, compared with 4.9% for non antibiotics, p<0.000).

Of 72 manufacturers, 30 (41.7%) made at least one of the substandard samples.

Most of the failures were clustered close to permissible quality limits, (depicted schematically in Fig 1 as the zone between the dotted lines). Fifteen samples (1.2%) from 11 manufacturers, deviated from permissible limits on assay or dissolution by 10% or more; only 1 of these was confirmed falsified by its market authorisation holder.

## Falsified medicines

No sample was deemed falsified based on test results. By December 2023, 45/79 market authorisation holders had checked sample details against manufacturing records, covering 578 samples; 21, from 12 companies, were confirmed falsified (3.6%). Seven of the falsified samples failed pharmacopeial tests (33.3%), compared with 9.4% of samples with confirmed correct production records (p <0.000).

20/21 falsified products were branded; six had fake batch numbers, while 15 had incorrect expiry dates. The majority of falsified samples (15/21) were purchased from unregulated internet sellers; only one was acquired from an outlet type permitted to sell prescription medicines. Study staff flagged 11/21 confirmed falsified products as suspicious at visual inspection.

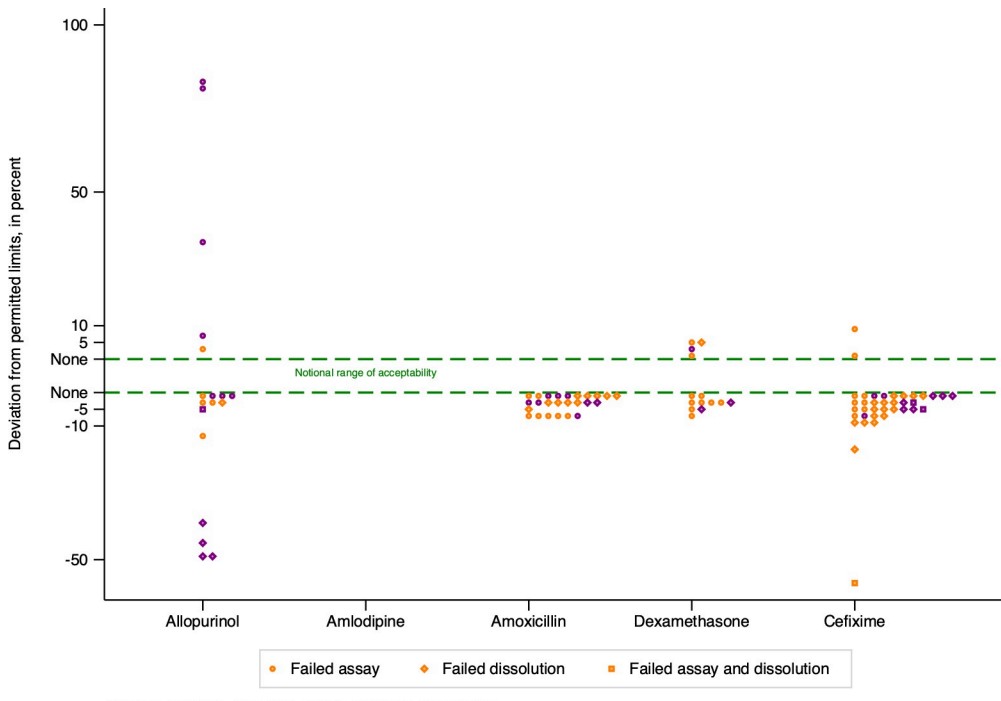

**Fig 1. For all samples failing assay or dissolution testing, maximum deviation from permitted limits, by active ingredient.**

## Source of medicines

In regression analysis, after controlling for differences in medicines, the odds of testing substandard were 2.2 times higher for samples collected in rural areas, compared with those collected from physical locations in cities (p<0.001). There was no significant difference in the pharmacopeial quality of samples acquired from different types of outlets. 8.0% of online medicines failed testing, against 8.3% acquired from physical sources (p = 0.83). However, among samples acquired online, those bought from individuals selling on general marketplaces or social media were more likely to fail any pharmacopeial test than those bought from licensed online vendors or verified online stores of physical pharmacies (11.3 vs 3.6%, p = 0.01).

Samples bought from the types of outlets included in BPOM's routine post-market surveillance were no more likely to fail testing than excluded outlets (7.8% vs 8.9%, p = 0.46).

## Branded status and price

Branded and unbranded products were equally likely to fail testing (9.1% vs 7.2%, p = 0.23) including after controlling for differences in medicine, district, or source. In unweighted analysis, medicines available free to patients in the public insurance system were significantly less likely to fail testing than medicines paid for out-of-pocket (3.8% vs 8.9%, p = 0.031).

Prices varied widely, both by brand and by outlet. In logistic regression, there was no relationship between price and pharmacopeial quality, including after controlling for differences in medicine, district, or source. The relationship between price and quality is reported in detail elsewhere [32].

## Weighted quality estimates

We collected at least one sample of two thirds of the registered medicines and doses specifically targeted in our sampling frame (for example amlodipine 5mg), as well a third of products of

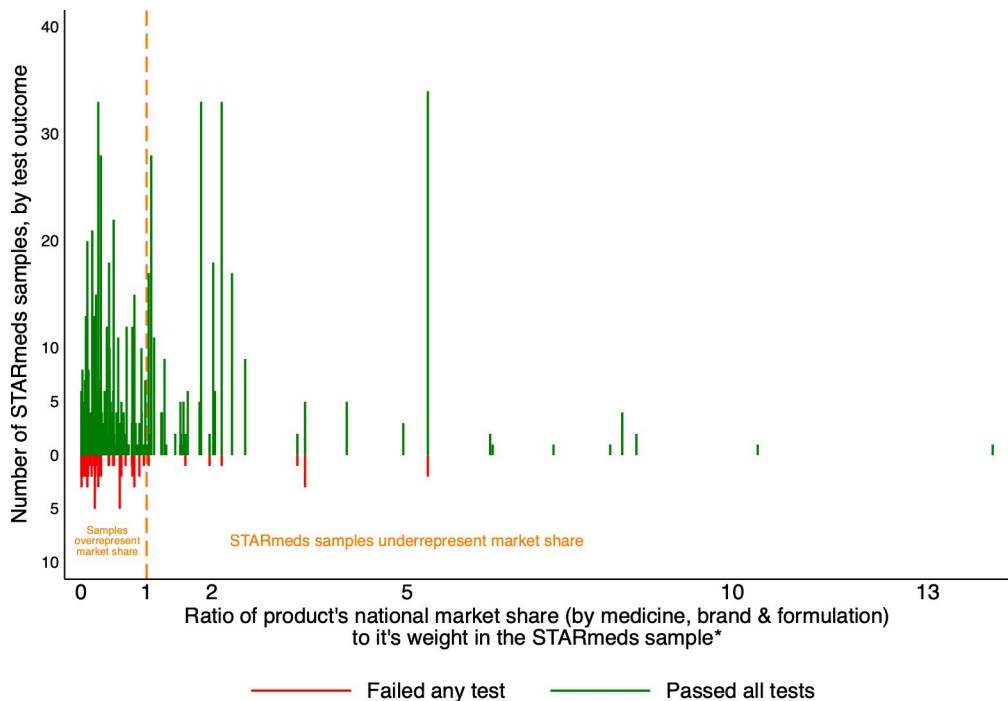

**Fig 2. STARmeds testing outcomes per product, plotted against ratio of weight in market to study weight.**
*Market share is calculated separately for the unregulated online market. Graph omits 4 samples with market share 16–76 times their share in the STARmeds sample; all passed testing.

other doses of the same formulations (e.g. amlodipine 10mg). In terms of market share, however, the samples we collected represented 97.1% of all targeted doses and 89.9% of other doses. Overall, we collected at least one sample of products representing 95.4% of the total Indonesian market for the five medicines and formulations included in STARmeds.

We weighted our survey results using the ratio of a brands' market share to its share of the total number of samples in our survey. Fig 2 plots the number of samples of each product tested (by pass/fail status) against its weight; S1 Table lists the respective market shares and sample shares, and gives weights.

Overall, 88% of the unique products in our sample (shown to the left of the dotted line in Fig 2) were over-represented, compared with their market share. This was partly because we deliberately sought unregulated online sellers who sell tiny volumes; the 14.6% of study samples from these sellers represented just 0.0022% of national market volume for study medicines. Additionally, we oversampled active ingredients with smaller distribution to allow for inter-medicine comparisons. For example, cefixime accounted for 15.3% of our sample, but just 3.7% of the market of our universe of 5 medicines. Failure rates were higher in products that were over-represented in our sample, compared with under-represented products (9.8% vs 3.3%, p<0.000).

As Fig 3 shows, adjusted prevalence of substandard samples across products tested in the study was 46.9% lower than raw prevalence (4.4% vs 8.2%). When we weighted samples by their relative market share within the universe of a single medicine (also shown in Fig 3), the prevalence of poor quality amlodipine fell by two thirds, while amoxicillin dropped by just 13%. While in unadjusted analysis public procurement medicines appeared less likely to fail testing compared with other medicines, the relationship was reversed, and lost statistical significance, once adjusted for the high volumes of public procurement medicines.

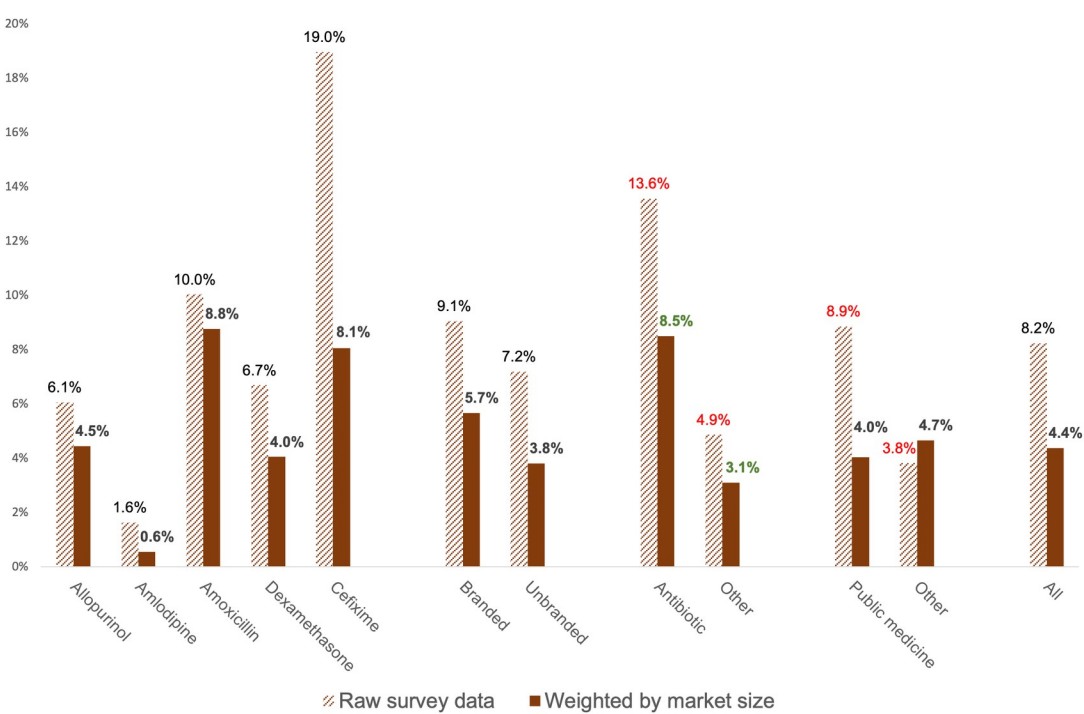

**Fig 3. Raw and adjusted prevalence for study medicines.** Red value labels denote statistically significant differences in analysis of raw data; green labels note differences significant in the adjusted analysis.

The weighted prevalence of medicines failing assay or dissolution by more than 10 percent of the permitted margin was 0.5%.

## Discussion

We collected 1,274 samples of five medicines across four therapeutic groups from all sources from which Indonesian patients commonly acquire medicines, and tested them for identity, quantification and uniformity of active ingredient, and dissolution. The raw prevalence of substandard products was 8.2%. However, when we adjusted for market size of individual brands, the estimate fell by 46.9%, to 4.4%. This suggests that poor quality medicines do not, in general, achieve large market shares. Using the raw rather than the adjusted estimate, we risk overestimating the volume of substandard study medicines sold in a year in Indonesia by 75 million units.

Our findings are consistent with the one other study we found which adjusted quality estimates for market size [33]. In that study of an antimalarial, an antibiotic and an analgesic in the Democratic Republic of Congo, the raw estimate of poor quality medicines (27.2%) fell to 1.3% when adjusted for market size data provided by distributors. (The authors acknowledge that, since samples were collected from distributors, these measures do not reflect degradation and other loss of quality in the supply chain.) Many academic studies of medicine quality included in recent metanalyses prioritise brand variety in their sampling [1, 5, 6, 34]. We therefore believe other, unadjusted studies may similarly mis-represent (and possibly overstate) the true proportion of medicines that are of poor quality.

Our adjusted estimate for five medicines was 10% higher than the 4.0% found by the Indonesian regulator BPOM in nation-wide post-market surveillance of over 10 times as many samples across all medicines in the months before the study [35]. STARmeds sampling strategy

and definitions of quality did not mirror BPOM's exactly; the differences could affect the results in both directions. We included outlets not licensed to sell prescription medicines, and used mystery shoppers rather than uniformed regulators to sample in retail outlets. Both factors could *increase* our chance of collecting poor quality products, compared with BPOM's strategy, although in fact, we found no significant difference in quality between samples collected from outlet types sampled by BPOM and others. (Such similarity between regulated and informal markets was also recent reported in Nigeria [30].) Other differences are more likely to *decrease* the relative chances of sampling poor quality products. We excluded sterile formulations at higher risk for production errors, and psychoactive medicines at greater risk for falsification, and did not include labelling errors in our definition of substandard products.

Despite these differences in definitions and sampling, the weighted estimate in our study is very similar to that published by the national regulator for the Indonesian market as a whole, yet it was achieved with a sample that was less than one tenth of the size. A full consideration of the cost of surveillance is published elsewhere [36], but we suggest that weighting raw estimates by market size is likely to be a resource-reducing option for arriving at a better understanding of the magnitude of the threat posed by substandard or falsified medicines in national markets. Sources of data on sales volumes vary by country. Data from commercial aggregators, where available, are expensive. Other potential sources include data from distributors, wholesalers, public procurers and public insurers, as well as tax and customs authorities.

Even the adjusted figure implies a large number of substandard products in the Indonesian market—close to 87 million tablets, capsules or 5ml doses across 5 medicines. However, most failed samples hovered close to Indonesia and the United States' permitted limits. Using one of the other 60 pharmacopeia used by national regulators [37], some may have met required thresholds. Since no pharmacopeia provides an evidence base for its chosen thresholds, it is impossible to estimate the clinical impact of samples deemed to have failed by a small margin. For many patients, especially in populations with lower body weights than those in standard-setting countries, it may be limited. We do not question the need for clear and well-enforced quality standards, but believe that a consistent and clear rationale for chosen pharmacopeial thresholds would help authorities prioritise measures of greatest importance to public health.

If quality is persistently low, the cumulative effect on a patient of even marginal deviations may be more severe, particularly in the case of therapy for chronic infections. Slightly lower than permitted dissolution of antibiotics may also place the product in the mutation selection window, favouring the development and spread of antibiotic-resistant bacteria [5, 38–40]. In our study, poor dissolution of cefixime is especially worrying, since WHO assigns it to the "Watch" category of antibiotics prone to resistance [41].

Our findings were comparable with other studies in Indonesia, Cambodia, India and Togo in finding that quality did not vary by price [42–44]. Researchers in Indonesia and other countries (including the United States, China and South Africa) have found that both patients and health-care providers distrust the quality of free medicines and unbranded generics; some suggest the distrust is deliberately sown to favour the financial interests of doctors, hospitals and pharmaceutical companies [21, 45–49]. Since the introduction of Indonesia's national health insurance scheme, Indonesian media have also carried reports questioning the quality of its free medicines [50, 51]. We found no basis for selective distrust of free medicines or unbranded generics in Indonesia.

We succeeded in verifying production records with 44/79 companies. Only one confirmed falsified product was found in a regulated outlet permitted to dispense prescription medicines, again mirroring a recent Nigerian study [30]. Unexpectedly, the majority of confirmed falsified samples passed quality testing; most had expiry dates which did not match genuine batch

numbers. This suggests criminals are extending dates on packaging of genuine products which continue to maintain serviceable quality beyond their authorised expiry dates.

Simple visual inspection by researchers familiar with brand packaging identified more than half of the products that were confirmed falsified. A pilot programme to train health care workers in Indonesia and Tanzania to spot and report suspect medicines was deemed promising and could be expanded [52]. However the signs would not be evident to most consumers, who do not have the same points of reference.

An important limitation of our study was that we could not afford to test for impurities. Even relatively well resourced regulators, including BPOM, do not commonly test for impurities, or for by-products of excipients; hence, the Indonesian regulator did not pick up the presence of lethal non-active ingredients in paediatric syrups during their extensive routine post-market surveillance in 2021/22 [53]. This underlines the need for more affordable medicine testing technologies, and greater attention to impurities and non-active ingredients in considering the quality of medicines [54–56].

## Conclusion

Although our study is one of the most comprehensive single-country academic surveys of medicine quality, we found that the "headline" figure of 8.2% substandard medicines it yielded likely overestimated both prevalence of poor quality products and their threat to public health. When adjusted for brand volumes, just 0.5% deviated from permitted limits by more than 10%. The overestimate was caused by sampling techniques which over-represented brands with small market share, and medicines bought from unregulated sources. Other published studies do not report relative market share. However, the diversity of brands and outlets they report within small sample sizes suggests that sampling techniques commonly oversample marginal brands which, in our study, were more likely to fail testing. Many also sample from unregulated outlets, where falsified medicines are most commonly found. In our study, we found that informal markets and unregulated internet sources represent only a small proportion of sales, again leading to over-representation of falsified products in the survey compared to the market. This may differ in other markets; we encourage systematic research to quantify the relative size of different market segments in other countries.

In common with authors of recent systematic reviews, we believe the aggregation of results across dissimilar studies is misleading [3, 5, 6, 34]. While quality-assurance will differ in countries with different regulatory capacity and medicine import profiles, our study suggests that such aggregations may compound the effect of sampling biases. Together with binary pass/fail reporting of medicines which fail testing only marginally, this has probably led to an overestimate of the potential health threat posed by substandard and falsified medicines globally.

We recommend that regulators and others conducting post-market surveillance or primary surveys of medicine quality consider weighting by market size to improve understanding of the volume and distribution of poor quality products, and to plan and resource effective responses. We note, however, that even estimates weighted by market volume are expensive and time-consuming to generate [36]. Health authorities must assess the value of this investment relative to a case-finding approach with narrower focus on identifying and removing substandard and falsified products at highest risk of harming public health [57], perhaps considering market-wide, volume weighted surveys of prevalence every four or five years, with more targeted, risk-based surveillance in interim years. We suggest this strategy could deliver a resource-effective balance of data necessary to plan and calibrate regulator responses and minimise risk to patients of substandard and falsified medicines.

## Supporting information

**S1 Methods. Detailed methods description for STARmeds study.**
(PDF)

**S1 Table. Market share and weighting of unique study products.**
(XLSX)

## Acknowledgments

STARmeds was a collaboration between Universitas Pancasila, Imperial College London and Erasmus University Rotterdam.

STARmeds study group members are listed alphabetically by institution. Group member roles for this paper are provided in the paper-level supplementary materials.

**Universitas Pancasila**

Yusi Anggriani, Esti Mulatsari, William Nathanial, Yunita Nugrahani, Elizabeth Pisani, Jenny Pontoan, Ayu Rahmawati, Mawaddati Rahmi, Stanley Saputra, Hesty Utami.

**Imperial College London**

Adrian Gheorghe, Katharina Hauck, Sarah Njenga, Sara Valente de Almeida.

**Erasmus University Rotterdam**

Amalia Hasnida.

The team would like to thank Indonesia's medicine regulator (Badan Pengawas Obat dan Makanan, BPOM) and the national statistics bureau (Badan Pusat Statistik, BPS) for active support in developing the methods described here. The majority of the data collectors were partners of BPS. We thank them for their hard work. We thank BPS statistical consultants Ardi Adji and Budi Santoso.

We also thank members of a multisectoral working group on medicine quality estimation known as PEMO, which groups 12 Indonesian government institutions and 5 professional or industry associations, for advice provided over the course of the study, as well as the members of our Study Advisory Group (Michael Deats, Kharisma Nugroho, Yodi Mahendradhata, Raffaella Ravinetto, Selma Siahaan, Val Snewin, Virginia Wiseman and Firman Witoelar) for helpful advice.

We thank United States Pharmacopeia, who provided reference standards at discounted prices.

## Author Contributions

**Conceptualization:** Elizabeth Pisani, Yusi Anggriani.

**Data curation:** Ayu Rahmawati, Mawaddati Rahmi.

**Formal analysis:** Elizabeth Pisani, Ayu Rahmawati, Esti Mulatsari.

**Funding acquisition:** Elizabeth Pisani, Yusi Anggriani.

**Investigation:** Elizabeth Pisani, Esti Mulatsari, Mawaddati Rahmi, William Nathanial.

**Methodology:** Elizabeth Pisani, Ayu Rahmawati, Esti Mulatsari.

**Project administration:** William Nathanial, Yusi Anggriani.

**Resources:** William Nathanial.

**Supervision:** Elizabeth Pisani, Yusi Anggriani.

**Validation:** Esti Mulatsari.

**Visualization:** William Nathanial.

**Writing – original draft:** Elizabeth Pisani.

**Writing – review & editing:** Ayu Rahmawati, Esti Mulatsari, Yusi Anggriani.

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
