## [Decision Letter · Decision Letter 0]

10 Oct 2024

PGPH-D-24-01444

A randomised survey of the quality of antibiotics and other essential medicines in Indonesia, with volume-adjusted estimates of the prevalence of substandard medicines.

Dear Dr. Pisani,

Thank you for submitting your manuscript to PLOS Global Public Health. After careful consideration, we feel that it has merit but does not fully meet PLOS Global Public Health’s publication criteria as it currently stands. Therefore, we invite you to submit a revised version of the manuscript that addresses the points raised during the review process.

Both reviewers acknowledge the importance of your research on the prevalence of substandard medications in Indonesia, commending the study design and unique methodological approach. However, they also identified several areas where the manuscript would benefit from clarification and improvement, especially in providing clearer conclusions and adding more detail to specific sections. Please see their detailed feedback below.

We look forward to receiving your revised manuscript.

Kind regards,

Elize Massard da Fonseca, Ph.D.

Academic Editor

Journal Requirements:

Additional Editor Comments (if provided):

Reviewers' comments:

Reviewer's Responses to Questions

**Comments to the Author**

1. Does this manuscript meet PLOS Global Public Health’s publication criteria? Is the manuscript technically sound, and do the data support the conclusions? The manuscript must describe methodologically and ethically rigorous research with conclusions that are appropriately drawn based on the data presented.

Reviewer #1: Yes

Reviewer #2: Yes

2. Has the statistical analysis been performed appropriately and rigorously?

Reviewer #1: Yes

Reviewer #2: Yes

3. Have the authors made all data underlying the findings in their manuscript fully available (please refer to the Data Availability Statement at the start of the manuscript PDF file)?

Reviewer #1: Yes

Reviewer #2: Yes

4. Is the manuscript presented in an intelligible fashion and written in standard English?

Reviewer #1: Yes

Reviewer #2: Yes

5. Review Comments to the Author

Reviewer #1: This is an important article about prevalence of substandard medications in middle and low income countries, with a focus on Indonesia. It has a good design. I recommend it for publication and make a few suggestions for improving the article.

The abstract lacks a conclusion and summary at the end. Suggest adding a clear conclusion for lay audiences.

Similarly, the conclusion isn't very conclusive. I suggest the authors look at it again with fresh eyes and explain to lay readers the prevalence of the substandard medicines and their interpretation. It's unclear whether they think this is a problem of public health importance or not, and what the total prevalence of substandard medicines is. I think they are suggesting this isn't a big public health problem, but I read it twice and i'm still confused.

Otherwise the article is well written and seems to have a unique and detailed methodological approach.

Reviewer #2: The paper is a clearly written report of a well-designed study to investigate the scale of falsified medicines in Indonesia, which uses a robust and well developed sampling frame, and a unique weighting procedure to weight by market share, which more accurately captures the real-world impact of falsified medicine.

The paper makes a strong contribution and I don’t have any major suggestions. I would suggest some small edits for clarity:

Abstract: ’46.9% lower than the raw estimate’– is this % decline to this level of accuracy and decimal points important for the abstract?

Line 83: “many patients buy medicines outside the public system”.- can you say more about what this might entail and how common is it for consumers to do this?

Line 103: ‘chosen based on public health importance’ – you could here reassure the reader that this was done by a committee/consensus of experts or stakeholders.

Line 116: ‘selecting target outlets randomly.’: you could explain here that this means across all types, not within types.

Line 119/table 2: it would be good to know within the sample frames, what were the shares of public system/private and other (including medicine shops etc)

Line 185: what does this n=467 refer to? And does the IQVIA data relate to Indonesia and even areas or Indonesia you sample?

6. PLOS authors have the option to publish the peer review history of their article (what does this mean?). If published, this will include your full peer review and any attached files.

**Do you want your identity to be public for this peer review?** For information about this choice, including consent withdrawal, please see our Privacy Policy.

Reviewer #1: No

Reviewer #2: No

---

## [Decision Letter · Decision Letter 1]

12 Nov 2024

A randomised survey of the quality of antibiotics and other essential medicines in Indonesia, with volume-adjusted estimates of the prevalence of substandard medicines.

PGPH-D-24-01444R1

Dear Dr. Pisani,

We are pleased to inform you that your manuscript 'A randomised survey of the quality of antibiotics and other essential medicines in Indonesia, with volume-adjusted estimates of the prevalence of substandard medicines.' has been provisionally accepted for publication in PLOS Global Public Health.

Best regards,

Elize Massard da Fonseca, Ph.D.

Academic Editor

Reviewer Comments (if any, and for reference):

Reviewer's Responses to Questions

**Comments to the Author**

1. If the authors have adequately addressed your comments raised in a previous round of review and you feel that this manuscript is now acceptable for publication, you may indicate that here to bypass the “Comments to the Author” section, enter your conflict of interest statement in the “Confidential to Editor” section, and submit your "Accept" recommendation.

Reviewer #1: All comments have been addressed

2. Does this manuscript meet PLOS Global Public Health’s publication criteria? Is the manuscript technically sound, and do the data support the conclusions? The manuscript must describe methodologically and ethically rigorous research with conclusions that are appropriately drawn based on the data presented.

Reviewer #1: Yes

3. Has the statistical analysis been performed appropriately and rigorously?

Reviewer #1: Yes

4. Have the authors made all data underlying the findings in their manuscript fully available (please refer to the Data Availability Statement at the start of the manuscript PDF file)?

Reviewer #1: Yes

5. Is the manuscript presented in an intelligible fashion and written in standard English?

Reviewer #1: Yes

6. Review Comments to the Author

Reviewer #1: Thank you for addressing the previous comments.

7. PLOS authors have the option to publish the peer review history of their article (what does this mean?). If published, this will include your full peer review and any attached files.

**Do you want your identity to be public for this peer review?** For information about this choice, including consent withdrawal, please see our Privacy Policy.

Reviewer #1: No
